# PTEN Overexpression Alters Autophagy Levels and Slows Sodium Arsenite-Induced Hepatic Stellate Cell Fibrosis

**DOI:** 10.3390/toxics11070578

**Published:** 2023-07-03

**Authors:** Fei Huang, Guanxin Ding, Yanjie Yuan, Lijun Zhao, Wenmeng Ding, Shunhua Wu

**Affiliations:** 1Department of Occupational and Environmental Health, College of Public Health, Xinjiang Medical University, No. 567 Shangde North Road, Shuimogou District, Urumqi 830011, China; 13565928062@163.com (F.H.); guanshaodgx@163.com (G.D.); zhaolijun2021@126.com (L.Z.); dingwenmeng@163.com (W.D.); 2The First Division Hospital of Xinjiang Production and Construction Corps, No. 4, Jiankang Road, Aksu City 843000, China; 13139938033@163.com

**Keywords:** inorganic arsenic, liver fibrosis, autophagy, Notch1/Hes1/PTEN

## Abstract

Exposure to inorganic arsenic remains a global public health problem. The liver is the main target organ, leading to arsenic-induced liver fibrosis. Phosphatase and tensin homology deleted on chromosome ten (PTEN) may participate in arsenic-induced liver fibrosis by regulating autophagy, but the exact mechanisms remain unclear. We established a mouse model of arsenic poisoning through their drinking water and a fibrosis model using the human hepatic stellate cell line LX-2 through NaAsO_2_ exposure for 24 h. Masson staining measured liver fibrosis. The cells were transfected with a PTEN overexpression plasmid. Western blot and qRT-PCR determined the levels of protein/mRNA expression. Fibrosis was evident in both the mouse model and arsenic-exposed LX-2 cells. NaAsO_2_ upregulated expression of autophagic markers microtubule-associated protein light chain A/B (LC3), recombinant human autophagy effector protein (Beclin-1), and hairy and enhancer of split homolog-1 (HES1), but downregulated PTEN. Alongside this, α-smooth muscle actin (α-SMA) expression was significantly upregulated by NaAsO_2_. PTEN overexpression altered NaAsO_2_-induced autophagy and downregulated LC3 and Beclin-1. While Notch1, HES1, α-SMA, and collagen I expression were all downregulated in the NaAsO_2_ groups. Therefore, PTEN overexpression might decrease autophagy and inhibit fibrosis progression caused by arsenic, and the NOTCH1/HES1 pathway is likely involved.

## 1. Introduction

Arsenic contamination is a serious environmental and geochemical issue endangering human health [1]. Exposure to inorganic arsenic (iAs) compounds causes not only skin lesions, kidney damage, and peripheral nervous system injury but also liver injury, liver fibrosis, cirrhosis, and liver cancer [2]. The liver is the most important target organ of arsenic metabolism [3]. Epidemiological studies linked chronic iAs exposure to an increased risk of liver disease, fibrosis, cirrhosis, and liver carcinogenesis [4,5,6]. The mechanisms underlying arsenic-induced liver fibrosis are multifaceted and include oxidative stress, inflammatory response, apoptosis, necrosis, and methylation [7,8,9]. In recent years, studies have demonstrated that autophagy plays a critical role in arsenic-related carcinogenic mechanisms.

Autophagy is essentially a protein degradation system involving the cell’s own lysosomes; it plays a vital role in removing misfolded proteins and damaged organelles to maintain cellular homeostasis [10,11]. Significantly, arsenic exposure affects autophagy in a dose-dependent manner. Multiple autophagy-related genes (ATGs) and signaling pathways co-regulate autophagy processes and biological functions [12]. As of September 2022, a total of 232 autophagy-related genes were obtained from the Human Autophagy Database (HADb), with the key signaling molecules being recombinant human autophagy effector protein (Beclin-1), VMP1, Atg5-Atg12, Atg4, and microtubule-associated protein light chain A/B (LC3), while the phosphatidylinositol 3 kinase/protein serine-threonine kinase (PI3K-Akt) pathways also play a role in autophagy [13].

Phosphatase and tensin homolog deleted on chromosome 10 (PTEN) is a tumor suppressor gene that exerts vital effects on cell growth, proliferation, migration, signal transmission, invasion, and apoptosis [14]. Existing studies have also demonstrated that abnormal PTEN expression is associated with non-neoplastic disease [15]. Activation of pro-fibrotic signaling pathways through PTEN may lead to fibrosis in the liver, lung, and kidney tissues [16,17]. Some studies have found that PTEN may be involved in the regulation of fibrosis in various organs through autophagy [18,19]. Xun Lai et al. [17] found that reduced autophagy enhances PTEN expression, while increased autophagy reduces PTEN expression. This suggests that degradation of PTEN by autophagy via the interaction of PTEN-p62 acts as a novel method of tumorigenesis involving highly up-regulated in liver cancer (HULC) [20]. Qing Yin et al. [21] in diabetic nephropathy found that miR-155-5p promoted autophagy and attenuated interstitial fibrosis by targeting PTEN. In addition, PTEN is involved in the regulation of multiple pathways. For example, PTEN negatively regulates phosphatidylinositol 3-kinase (PI3K) involved in the regulation of the PI3K-Akt-mTOR pathway [22]. Hairy and enhancer of split 1(Hes1) can affect the transcription level of PTEN mRNA and downregulate the expression of PTEN protein, thereby affecting the Notch1-HES1-PTEN pathway [23]. However, the exact mechanism underlying the interaction of PTEN, autophagy, and arsenic-induced liver fibrosis is not fully understood.

This study assessed mouse response to iAs^3+^, with histological changes suggesting liver fibrosis and changes in liver protein expression. The results were confirmed in vitro, with human hepatic stellate cell (HSC) responses to different iAs^3+^ doses to characterize the full range of chronic arsenic exposure levels. Then, we investigated overexpression of PTEN during iAs^3+^-induced fibrosis and explored the effect of the NOTCH1/HES1/PTEN signaling pathway on autophagy. Our data could be used to explore changes in autophagy-related proteins during iAs-induced fibrosis in human HSCs. These results could lay a foundation and provide new insights for further liver fibrosis investigations. These results could be important in areas where arsenic exposure through drinking water and food is significant.

## 2. Materials and Methods

### 2.1. Animals and Treatments

Twenty-four healthy sterile C57BL/6 male mice (weighing 20–22 g) were purchased from the Animal Experiment Center of Xinjiang Medical University (Xinjiang, China) and maintained on 24 h adaptive feeding for 1 week at a temperature of 25 ± 2 °C and a relative humidity of 45 ± 5%. The mice were divided into four groups of six each using a random-number table. The LD50 of sodium arsenite (iAs^3+^, analytical grade, No. 3 Chemical Reagent Factory, Beijing, China) in the mice was 16.2 mg/kg, as determined by the Horn method [24] in a preliminary experiment. In brief, the Horn method involves a preliminary test and a formal test. The preliminary test determined the exposure dose of the test substance according to the relevant toxicological data of the similar structural chemical substances. The formal test dose series was then determined according to the results of the preliminary test. The animals were randomly divided into 4 groups, with 4 animals in each group. The sodium arsenite was diluted with distilled water in different doses, and administered orally at 0.01 mL/g. After exposure, the animals were closely observed and recorded in detail with regard to abnormal behavior, the number of animals that died, and the time of death. The observation period of all infected animals was 14 days. We administered 1/15 of the LD50 as a dose in the high-dose group, and the differences among the dose groups were twofold: high-dose iAs^3+^ group (H), NaAsO_2_ 1.08 mg/kg; low-dose iAs^3+^ group (L), NaAsO_2_ 0.54 mg/kg; the animals in the normal control group I drank deionized water solution for 24 weeks. During the exposure period, the water was freshly prepared and recorded daily, and the animals were weighed three times a week. We weighed the sodium arsenite powder and diluted it into a liquid with deionized water. To prevent oxidation of the sodium arsenite, we fed the water to the mice immediately after preparation. The mice were fed the water only once a day to reduce the exposure of the water to air. The mice were fed in groups for 4, 8, 16, and 24 weeks before they were killed. All animal experiments were granted ethics committee approval and were conducted in accordance with the regulations of the Ethics Committee of Xinjiang Medical College and the Guidelines for the Care and Use of Laboratory Animals of the Chinese National Institute of Health (Ethical approval number: IACUC-20210309-08).

### 2.2. Masson Staining for Detecting Liver Fibrosis

Fresh liver tissues isolated from C57BL/6 mice were fixed in 10% formalin (Beijing Solaibao Technology Co., Beijing, China), dehydrated, embedded in paraffin (Leica, Wetzlar, Germany), and sliced. Then, liver sections (4–5 μm) were dewaxed and rehydrated, and according to the standard procedure of the Masson staining kit (Solarbio, Beijing, China), tissue staining was performed. After being sealed with gum, the sections were observed and photographed using an optical microscope.

### 2.3. Cell Culture

LX-2 cells (Punosai Life Technology Co., Wuhan, China) were cultured at 37 °C in an atmosphere containing 5% CO_2_ and 95% air at 100% humidity in Dulbecco’s modified eagle medium (DMEM) (American Hyclone Co., Logan City, UT, USA) plus 10% heat-inactivated fetal bovine serum (FBS) (American Hyclone Co., Logan City, UT, USA), 100 mg mL^−1^ penicillin, and 100 mg mL^−1^ streptomycin. When cells reached 80–90% confluence, they were trypsinized in 0.25% trypsin and placed in 6-well plates for iAs trioxide exposure. The concentration of iAs trioxide exposure for the cell groups is shown in Table 1.

### 2.4. Transmission Electronic Microscopy

LX-2 cells were fixed with 4% glutaraldehyde and 1% osmium tetroxide, rinsed in 100 mM sodium phosphate buffer, dehydrated in ethanol, and embedded in EPON. Ultrathin sections of LX-2 cell were collected on formvar-coated grids and stained with 10% uranyl acetate and 1% lead citrate; then, the ultrastructure of autophagosomes was examined using a JEM100CXII transmission electronic microscope (TEM) (Hitachi, Tokyo, Japan) operated at 80 kV.

### 2.5. Cell Transfection

The LX-2 cells were transfected with a PTEN overexpression plasmid and control vector plasmid (Jikai Gene Biology Co., Shanghai, China). LX-2 cells (Punosai Life Technology Co., Wuhan, China) were cultured at 37 °C in 5% CO_2_ and 95% air at 100% humidity in DMEM plus 10% heat-inactivated FBS, 100 mg mL^−1^ penicillin, and 100 mg mL^−1^ streptomycin. The medium was changed once every 2 days. Transfection was carried out using a six-well plate until the growth of the cells reached 70–80%. A volume of 500 μL of DMEM culture medium without antibiotics and serum was transferred into a 1.5-mL sterile centrifuge tube. Then, we added 5 μg plasmid DNA and 8 μL polyethylenimine (PEI), followed by mixing and shaking. Further, the mixture was left to stand for 10 min at room temperature, followed by the addition of 1.5 mL of DMEM medium without antibody and serum to the well plate. When the reaction time was completed, the transfection reagent and a DNA mixture were added to the whole well in a uniform drop at 500 uL per well and were then gently mixed. To achieve the highest transfection efficiency, cells had to be replaced with fresh complete culture medium after 4–6 h of post-transfection incubation.

### 2.6. Quantitative Real-Time PCR (qRT-PCR Analysis)

Total RNA was isolated from LX-2 cells and liver tissues using the RNAiso kit (TianGen, China) following the manufacturer’s instructions. Total RNA was reverse-transcribed to cDNA using the Primer Script™ RT Master Mix kit (Takara, Japan). Then, qRT-PCR was performed using the SYBR ^®^ Premix Ex Taq™II (2×) mix (Takara, Japan) using a CFX 96-type RT fluorescence quantitative PCR instrument (Bio-Rad, Hercules, CA, USA). GAPDH was used as a reference gene. Cycle thresholds were determined for each sample and each gene amplification. Based on the 2^−ΔΔCt^ method, relative target gene expression was calculated. Primer sequences for the mouse and cell-based experiments are shown in Table 2. The PCR procedure was an initial 30 s hold at 95 °C, then 40 repeated cycles of step 1 at 95 °C for 5 s, step 2 at 60 °C for 20 s, and step 3, melting (65–95 °C depending on the primer melting temperature) at 2.2 °C/s.

### 2.7. Western Blot

Total protein was extracted from cells and liver tissues using ice-cold RIPA lysis buffer (plus phenylmethylsulfonyl fluoride). Next, the total protein concentrations were determined by a bicinchoninic acid assay (BCA) protein quantification kit (Thermo Scientific, Waltham, MA, USA). Protein samples (30 μg) were separated using sodium dodecyl sulfate-polyacrylamide gel electrophoresis (SDS-PAGE) with electrophoresis solution (Solarbio Co., Beijing, China). The proteins were further transferred to 0.22-μm polyvinylidene fluoride membranes (Sangon Biotech, Shanghai, China) with electrotransfer solution (Solarbio Co., Beijing, China). After blocking in 5% skimmed milk for 1 h at 37 °C, the blots were incubated overnight at 4 °C with the following primary antibodies: LC3 (1:1000) (Proteintech, Rosemont, IL, USA, 14600-1-AP), Beclin-1 (dilution 1:2000) (Bioss, Woburn, MA, USA, bs-1353R), α-smooth muscle actin (α-SMA) (1:1000) (Abcam Co., Cambridge, UK), Collagen I (1:1000) (Abcam Co., UK, ab124964), HES1 (1:1000) (Abcam Co, UK), PTEN (1:1000) (CST, 11988S), NOTCH1 (1:500) (Abcam Co., UK, ab52627), GAPDH (1:10,000) (Abcam Co., UK), and β-actin (1:10,000) (Abcam Co., UK). The blots were rinsed three times with Tween (Sangon Biotech) plus PBS and incubated with horseradish peroxidase (HRP)-conjugated secondary antibodies (Sangon Biotech) (1:8000) for 1h. Immunoreactivity was visualized using an alkaline phosphatase color development kit (Sangon Biotech). Image analysis software ImageJ v1.51 (National Institutes of Health, Bethesda, MD, USA) was employed to calculate relative protein expression.

### 2.8. Statistical Analysis

SPSS 25.0 software (IBM Corp., Armonk, NY, USA) was utilized for statistical analysis. The quantitative data that were normally distributed were expressed as mean ± standard deviation (x ± s). One-way ANOVA was applied for comparison between groups, and LSD was used for further pairwise comparisons. The median and interquartile spacing [M (QR)] were implemented to describe the non-normal distribution, and the Kruskal–Wallis H test was used for comparison between groups.

## 3. Results

### 3.1. Autophagy and Fibrosis Were Established in the Liver Tissues of Mice Exposed to Different Concentrations of Sodium Arsenite

The HE staining of mouse liver tissue showed that the hepatocytes in the control group were uniform in size and morphology, with hepatocytes centered on the central vein and extending radially to the periphery. The hepatic cords were neatly arranged, with no degeneration, necrosis, inflammatory cell infiltration, and fibrous tissue proliferation in the intact liver lobules, and no hepatocyte degeneration. At 4 weeks of sodium arsenite exposure, aqueous degeneration and steatosis were observed in the hepatocytes in the low-dose group, with occasional inflammatory cell infiltration, but the structure of the liver lobules was still intact. In the high-dose group, hepatocyte steatosis was observed in the liver tissue, with obvious interspersed distribution of lipid droplets, scattered punctate and sheet necrosis, and local inflammatory cell infiltration in the visual field. In the high-dose group, the pathological changes were significantly aggravated. Pseudo-lobules were formed in the hepatic portal area segmented by the fibrous septum, with obvious punctal necrosis and abnormal proliferation of the hepatocytes, as well as edema and inflammatory cell infiltration (Figure 1).

Masson staining verified the establishment of a liver tissue fibrosis model in mice induced by sodium arsenite. As shown in the figure, the normal group had only a small amount of blue-stained collagen fibers distributed in the portal area, without obvious collagen hyperplasia. For short-term (24 h) arsenic exposure, a small amount of vacuoles formed by necrotic liver cells could be seen in the low-dose sodium arsenite group. The blue-stained fibrous tissue extended outward from the portal area or the central vein; in the high-dose sodium arsenite group, the wall of the central vein was thickened, and a large number of blue-stained collagen fibers appeared and surrounded the portal area (Figure 2).

Transmission electron microscope analysis of LX-2 cells treated with iAs^3+^ showed that the hepatocytes in the control group were normal and irregular. The nucleus was round or oval, with a clear nuclear envelope and nucleolus. Organelles, such as the mitochondria, the endoplasmic reticulum, and the ribosomes all showed normal morphology and were scattered within the cytoplasm. Compared with the control group, t“e “false Dirichlet ”ap” could be seen between the LX-2 cells in the iAs^3+^-dosed cells, and with high-dose iAs^3+^, the gap was widened. Compared with the control group, the number of autophagosomes in the high-dose iAs^3+^ group was significantly increased after 48 h of treatment (Figure 3).

### 3.2. PTEN Is Involved in the Autophagy and Fibrosis in Mouse Liver Tissue Caused by Sodium Arsenite Treatment

In order to determine whether PTEN was involved in iAs^3+^-induced autophagy and fibrosis in the mouse liver tissue, we examined LC3, Beclin-1, PTEN, α-SMA, and HES1 mRNA expression levels in liver tissues from the mice after exposure to NaAsO_2_.

When we compared the control and iAs^3+^ treatment groups, LC3 mRNA expression was significantly higher than that of the control in both the low- and high-dose iAs^3+^ groups at 24 weeks (*p* < 0.05); Beclin1 expression was significantly higher in the low-dose iAs^3+^ group than in the control group (*p* < 0.05). Similar to Beclin-1, α-SMA mRNA expression levels increased in the low-dose iAs^3+^ group. In all treatment groups, PTEN mRNA expression levels tended to initially increase and then decrease with increasing As^3+^ doses (low iAs^3+^: 1.79 ± 0.17; medium iAs^3+^: 0.84 ± 0.10; *p* < 0.001) (Table 3).

At 24 weeks, the LC3 and Beclin-1 protein expression level varied among the treatment groups (*p* < 0.05) (Table 4, Figure 1). LC3 protein expression in the high-dose iAs^3+^ group was lower than that in the control group (*p* < 0.05). Compared with the control group, the expression level of Beclin-1 protein in the low-dose iAs^3+^ group and high-dose iAs^3+^ group was significantly increased, and the difference was statistically significant (*p* < 0.05). PTEN protein expression in the high-dose iAs^3+^ group was lower than that in the control group at 24 weeks (*p* < 0.05). HES1 protein expression in the low-dose iAs^3+^ group was higher than that in the control group (*p* < 0.05). Comparison among each dose group showed that the α-SMA protein expression level increased with prolonged exposure time, and the difference compared with the control group was statistically significant at 24 weeks (*p* < 0.05). Under different doses at same time points, PTEN mRNA and protein expression levels were negatively correlated with dose (Table 4, Figure 4).

### 3.3. PTEN Overexpression Can Reduce Autophagy Induced by Sodium Arsenite

To confirm that PTEN was involved and identify a relationship between autophagy, we then investigated the overexpression of PTEN in LX-2 cells after iAs^3+^ treatments. Compared with the control, both the blank plasmid and high-dose iAs^3+^ + blank plasmid group showed decreased PTEN mRNA levels. In the PTEN overexpression group, compared with high-dose iAs^3+^ + blank plasmid group, PTEN mRNA levels were increased (*p* < 0.05) (Table 5).

The effects If iAs^3+^ on LC3 and Beclin-1 protein expression in LX-2 cells after plasmid infection were analyzed. When we compared the blank plasmid and high-dose iAs^3+^ + blank plasmid group, LC3 and Beclin-1 protein expression levels were significantly increased with sodium arsenite treatment (*p* < 0.05). In the PTEN overexpression group, compared with the high-dose iAs^3+^ + blank plasmid group, LC3 and Beclin-1 protein expression levels were significantly decreased (*p* < 0.05) (Table 5, Figure 5).

The protein expression level of Notch1 in the high-dose iAs^3+^+ blank plasmid group was significantly increased (*p* < 0.05) in comparison with that in the control group. Furthermore, the expression level of Notch1 protein in the PTEN overexpression group was significantly downregulated (*p* < 0.05) as compared with that in the high-dose iAs^3+^ + blank plasmid group. The expression level of HES1 protein in the high-dose iAs^3+^ + blank plasmid group was significantly higher (*p* < 0.05) than that of the control group. Compared with the high-dose iAs^3+^ + blank plasmid group, the expression level of HES1 protein in the PTEN overexpression group was significantly downregulated (*p* < 0.05) (Figure 5 and Table 5).

### 3.4. PTEN Overexpression Can Reduce Fibrosis Induced by Sodium Arsenite in LX-2 Cells

In order to examine fibrosis in the cell-based system, we examined α-SMA and collagen I protein expression in LX-2 cells. After sodium arsenite exposure, the expression level of α-SMA protein in the high-dose iAs^3+^ + blank plasmid group was significantly higher (*p* < 0.05) than that in the control group. Compared with the high-dose iAs^3+^ + blank plasmid group, the expression level of α-SMA protein in the PTEN overexpression group was significantly downregulated (*p* < 0.05). The protein expression level of collagen I in the high-dose iAs^3+^ + blank plasmid group was significantly higher (*p* < 0.05) than that in the control group. As visible in Figure 6 and Table 6, the protein expression level of collagen I in the PTEN overexpression group was significantly downregulated (*p* < 0.05) compared with that in the high-dose iAs^3+^ + blank plasmid group (Figure 6 and Table 6).

## 4. Discussion

This study aimed to explore the effects of PTEN with NaAsO_2_ on liver fibrosis and autophagy. We preliminarily found that the abnormal expression of PTEN was correlated with autophagy in the process of liver fibrosis induced by sodium arsenite in mice. The overexpression of PTEN reduced the occurrence of autophagy and slowed down the liver fibrosis caused by sodium arsenite. NOTCH1/HES1/PTEN might be involved in the regulation and affect the occurrence of autophagy and fibrosis. This study, therefore, identified new targets for clinical prevention of arsenic poisoning and alleviation of side effects of arsenic in combination therapy of tumor chemotherapy.

In this study, we established an arsenic poisoning model in mice. Initially, the mice were divided into four groups: normal, low-dose group, medium-dose group, and high-dose group. However, since there was no significant difference in the low-dose group, it was removed from the experiment, resulting in three groups. To ensure accuracy, we consulted relevant literature and sought guidance from the experimental instructor in the animal housing room. Considering that liver fibrosis is a chronic change rather than acute death in a short period of time, the setting of IC50 is deemed unnecessary. The assessment of fibrosis changes primarily relies on pathological results. HE and Masson staining indicated liver fibrosis in the models. In both of them, arsenic exposure led to significant liver fibrosis, with an upregulated expression of collagen I and α-SMA. Epidemiological investigations reveal links between arsenicosis and various malignant tumors, such as lung cancer, skin cancer, liver cancer, gallbladder and gastrointestinal tumors, and lymphoma [25,26,27]. Liver inflammation due to inorganic arsenic can result in liver fibrosis, leading to cirrhosis and carcinogenesis [28]. HSCs are the key effector cells mediating the occurrence and development of liver fibrosis. HSC activation is mainly characterized by fibroblast proliferation, excessive collagen synthesis, extracellular matrix deposition (including Collagen I and Collagen III), and overexpression of α-SMA [29].

In our study, we also observed autophagosomes in the fibrosis model by TEM and found that with increased infection time and dose, the interstitial space of the mouse hepatocytes widened significantly, and the number of lipid droplets and autophagic vesicles with double-layered membranes were increased. Based on our data, in both animal and cellular experiments, both at the genetic level and at the translational level, it was shown that LC3 and Beclin1 positively correlated with autophagic activity with increasing dose and duration of sodium iAs^3+^ exposure. In recent years, autophagy has been associated with key roles in several human diseases. Autophagy maintains cellular homeostasis by regulating various physiological processes, including cytokine formation, pathogen clearance, antigen presentation, inflammatory responses, and innate and adaptive immune responses [30]. Autophagy activity and biological functions are regulated by several ATGs and associated proteins, including LC3, SQSTM-1/P62, Beclin-1, ATG4, ATG5, and ATG8. LC3 is involved in the formation of autophagosomes and dissociates from lysosomal structures to digest damaged materials. Beclin1 is a key regulator of autophagy and an inactive or dysfunctional Beclin1 leads to a suppressed autophagic process [31].

Our preliminary experiments revealed a dose-dependent relationship between the increase in arsenic concentration and the change in fibrosis observed in LX-2 cells, the most noticeable difference was observed between the high-dose group and the control group. Hence, we opted for plasmid transfection in the high-dose group. In this study, after the PTEN overexpression plasmid was transferred into human HSCs, the level of liver fibrosis and autophagy was downregulated. As an important regulator of liver fibrosis, PTEN participates extensively in the process of liver fibrosis by regulating the activities of hepatocytes, hepatic stellate cells, and macrophages. Low expression or loss of PTEN was previously observed in the fibrotic liver tissues of rats treated with activated hematopoietic stem cells and CCl4 [32]. Targeting PTEN alleviated liver fibrosis, while saponin A promoted the expression of PTEN by binding with DNMT1, thereby reducing liver fibrosis [33]. Bueno et al. [34] also found that after knocking out the PTEN gene in alveolar epithelial cells, the degree of pulmonary fibrosis was aggravated. Additionally, PTEN is associated with autophagy. Research indicates that SLC9A3R1 increases the expression of PTEN via interaction with PTEN, and PTEN increases autophagy, whereas the loss of PTEN results in the inhibition of autophagy [35]. In addition, inhibition of autophagy increases PTEN, whereas induction of autophagy decreases PTEN [36]. Therefore, the results of our study are consistent with these studies. This suggests that the PTEN gene may be an important target to alleviate the progression of fibrosis, and interfering with the expression of PTEN might help reduce fibrosis. However, the main difference from these earlier studies is that we found that when PTEN overexpression enhanced the occurrence of autophagy, the increased level of autophagy further contributed to the remission of fibrotic lesions. Therefore, overexpression of PTEN can inhibit autophagy during autophagosome formation and maturation, but autophagy does not inhibit the response to ATG binding. This dual role of PTEN in enhancing the degree of autophagy and alleviating liver fibrosis deserves more scientific attention.

We also found that Notch1/HES1 may be involved in sodium arsenite exposure-induced liver fibrosis and autophagy, and PTEN can regulate Notch1/HES1 to affect autophagy and fibrosis. Liu et al. [37] found that Notch1 regulated the expression of PTEN, inhibited autophagy through interaction with Hes1, and aggravated renal tubulointerstitial fibrosis in diabetic nephropathy. Our data showed that compared with the high-dose iAs^3+^ + blank plasmid group, the protein expression levels of Notch1 and HES1 in the PTEN overexpression group were significantly down-regulated. In addition, we used Illumina Human Methylation 850K genome-wide methylation microarray for detection of fibrosis in the stained hepatic stellate cells, in which we utilized high concentrations of sodium arsenite, as well as in the autophagy model group. Using these analyses, we aim to detect differentially methylated genes at the epigenetic level and further investigate the mechanism of PTEN involvement in the regulation of arsenic-induced liver fibrosis and autophagy. 

## 5. Conclusions

The results of this study showed that abnormal expression of PTEN was correlated with autophagy during liver fibrosis induced by sodium arsenite in mice. Similar results were found in HSCs, where overexpression of PTEN reduced the occurrence of autophagy and slowed down the liver fibrosis. The NOTCH1/HES1 signaling pathway might be involved in the regulation and affected the occurrence of autophagy and fibrosis. Most previous studies did not investigate PTEN and arsenic-induced liver fibrosis specifically. The present study highlights the importance of targeting PTEN for sodium arsenite-induced liver fibrosis and autophagy. Further research on the mechanistic issues of HSC management of arsenic exposure is warranted.

## Figures and Tables

**Figure 1 toxics-11-00578-f001:**
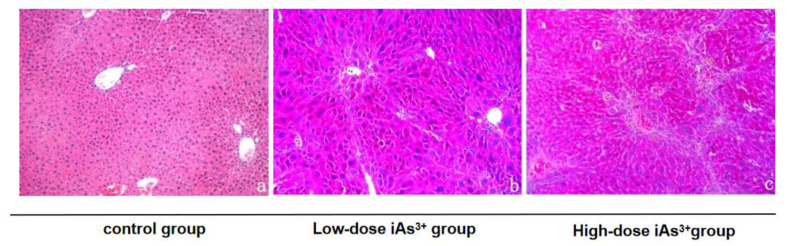
Effect of sodium arsenite exposure on liver histopathological changes in mice (H&E staining, ×100). Note: (**a**–**c**) represent the control group, low-dose sodium arsenite group, and high-dose sodium arsenite group, respectively.

**Figure 2 toxics-11-00578-f002:**
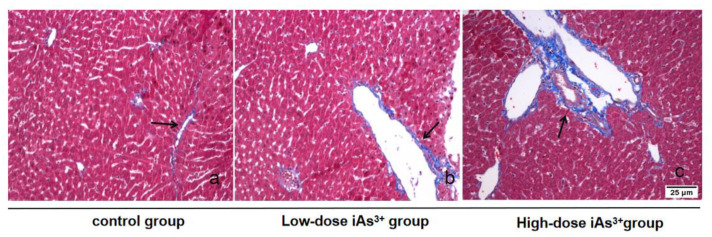
Masson staining verifies the formation of liver tissue fibrosis in mice caused by sodium arsenite (Masson, ×200). Note: (**a**–**c**) represent the control group, low-dose sodium arsenite group, and high-dose sodium arsenite group, respectively.

**Figure 3 toxics-11-00578-f003:**
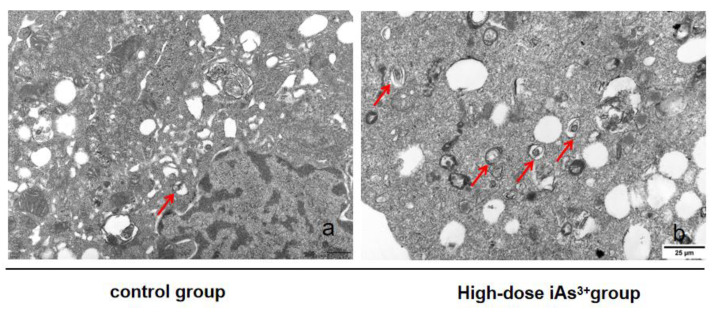
Transmission electron microscope observation of autophagosomes in LX-2 cells in each group (×100). Note: (**a**,**b**) represent the control group and the high-dose sodium arsenite group respectively. The arrow indicates the autophagosomes.

**Figure 4 toxics-11-00578-f004:**
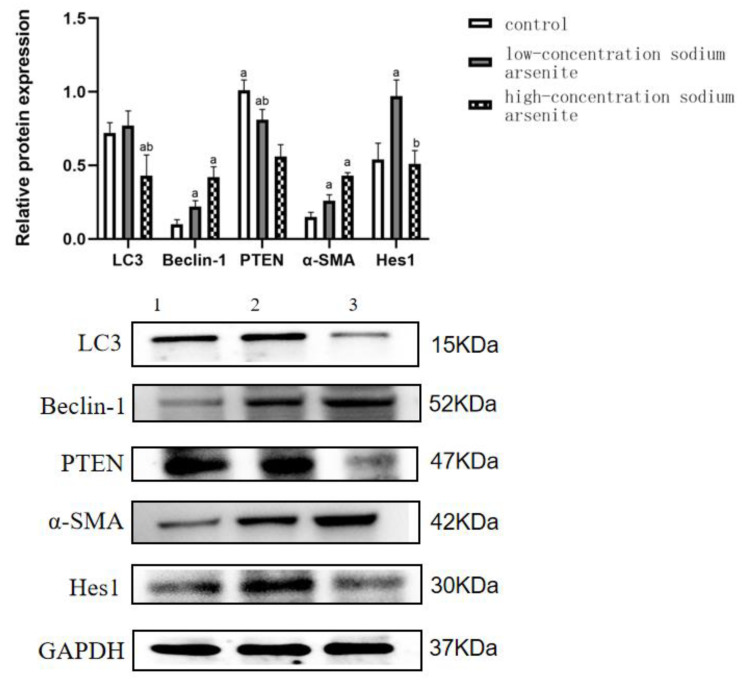
Effects of PTEN, LC3, Beclin-1, α-SMA, and HES1 protein expression in hepatic fibrosis in mice caused by sodium arsenite. Note: ^a^
*p* < 0.05, compared with the control group; ^b^
*p* < 0.05, compared with the low arsenic group.

**Figure 5 toxics-11-00578-f005:**
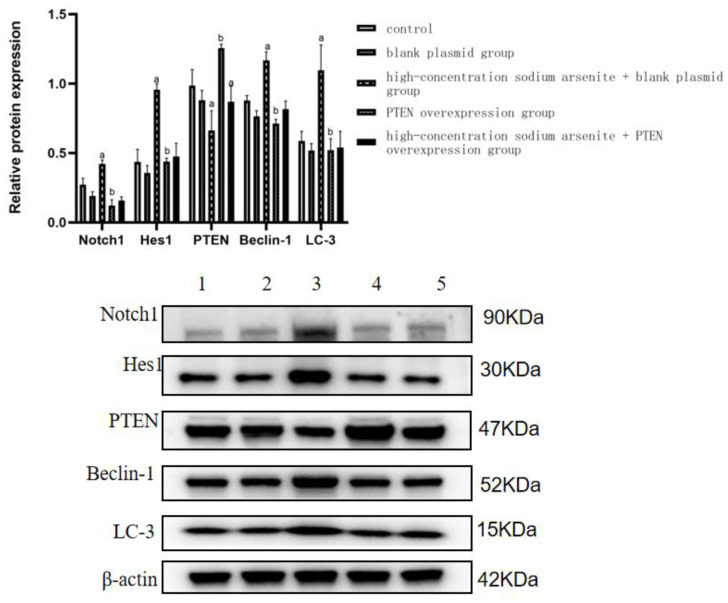
Effect of sodium arsenite PTEN overexpression on Notch1/HES1/PTEN and autophagy-related protein expression in LX-2 cell fibrosis by sodium arsenite. (Note: 1–5 are the control group, PTEN blank plasmid group, high-dose sodium arsenite group + blank plasmid group, PTEN overexpression group, and high-dose sodium arsenite group + PTEN overexpression group, respectively. Compared with the control group, ^a^ *p* < 0.05; compared with the high-dose sodium arsenite group + blank plasmid group, ^b^
*p* < 0.05).

**Figure 6 toxics-11-00578-f006:**
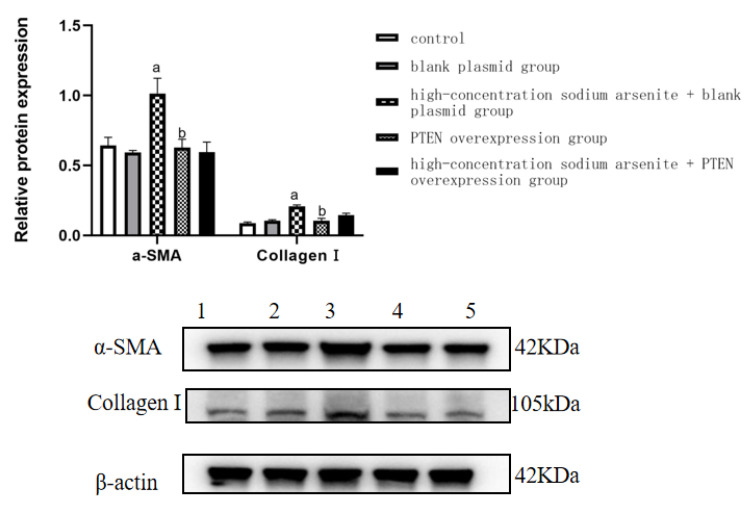
Effect of sodium arsenite on the expression of α-SMA and Collagen I proteins in LX-2 cells. (Note: 1–5 are the control group, PTEN blank plasmid group, high-dose sodium arsenite group + blank plasmid group, PTEN overexpression group, and high-dose sodium arsenite group + PTEN overexpression group, respectively. Compared with the control group, ^a^
*p* < 0.05; compared with the high-dose sodium arsenite group + blank plasmid group, ^b^
*p* < 0.05).

**Table 1 toxics-11-00578-t001:** Cell groups.

Group	Arsenic Concentration
High-dose iAs^3+^ group	5.00 μmol/L
Middle-dose iAs^3+^ group	0.50 μmol/L
Low-dose iAs^3+^ group	0.05 μmol/L
Control group	0.00 μmol/L

**Table 2 toxics-11-00578-t002:** Primer sequences.

Gene	Forward (5′–3′)	Reverse (3′–5′)
α-SMA	GCGTGGCTATTCCTTCGTGACTAC	CGTCAGGCAGTTCGTAGCTCTTC
LC3	GAACAGGAGAAGGATGAAGACGG	AATCCACTGGGGACTGAAATAGC
Beclin-1	CCAGGAACTCACAGCTCCATTAC	GGATGAATCTTCGAGAGACACCA
PTEN	GCTGAGAGACATTATGACACCGC	TTACACCAGTCCGTCCCTTTC
HES1	TGGAGAGGCTGCCAAGGTTT	ACTGGAAGGTGACACTGCGTT
GAPDH	CGTGTTCCTACCCCCAATGTG	TAGCCCAAGATGCCCTTCAGT

**Table 3 toxics-11-00578-t003:** Effect of the expression of PTEN, LC3, Beclin-1, α-SMA, and HES1 mRNA in mouse liver fibrosis induced by sodium arsenite.

Group	LC3	Beclin-1	PTEN	α-SMA	HES1
Control	1.04 ± 0.05	1.03 ± 0.03	1.03 ± 0.04	1.02 ± 0.02	1.03 ± 0.08
Low-dose iAs^3+^ group	2.66 ± 0.37 ^a^	1.84 ± 0.17 ^a^	1.79 ± 0.17 ^a^	3.05 ± 0.52 ^a^	1.49 ± 0.22 ^a^
High-dose iAs^3+^ group	3.25 ± 0.29 ^ab^	1.09 ± 0.10 ^b^	0.84 ± 0.10 ^b^	1.12 ± 0.13 ^b^	1.09 ± 0.70
*F*	54.111	47.218	56.167	41.351	9.951
*p*	<0.001	<0.001	<0.001	<0.001	0.012

Note: The values given are relative expression levels from the PCR analysis. ^a^ indicates a significant difference (*p* < 0.05) between the value and the control group; ^b^ indicates a significant difference (*p* < 0.05) between the value and the low-dose iAs^3+^ group. The *p*-value is a measure of the difference between the control group and the experimental group; The *F* test was used to assess the significance of the mean difference between two or more samples. The *F* value represents the ratio of two mean squares (effect item/error item), with no negative value possible. The greater the *F* value, the more obvious the difference, and the higher the test accuracy.

**Table 4 toxics-11-00578-t004:** Effects of the protein expression of PTEN, LC3, Beclin-1, α-SMA, and Hes 1 in mouse liver fibrosis induced by sodium arsenite.

Group	LC3	Beclin-1	PTEN	α-SMA	HES1
Control	0.72 ± 0.07	0.10 ± 0.03	1.01 ± 0.07	0.15 ± 0.03	0.54 ± 0.11
Low-dose iAs^3+^ group	0.77 ± 0.10	0.22 ± 0.04 ^a^	0.81 ± 0.07 ^a^	0.26 ± 0.04 ^a^	0.97 ± 0.11 ^a^
High-dose iAs^3+^ group	0.43 ± 0.14 ^ab^	0.42 ± 0.07 ^a^	0.56 ± 0.08 ^ab^	0.43 ± 0.02 ^a^	0.51 ± 0.09 ^b^
*F*	8.809	33.706	24.849	69.346	17.210
*p*	0.016	0.001	0.003	<0.001	0.003

Note: The values given are relative expression levels from the western blot analysis. ^a^ indicates a significant difference (*p* < 0.05) between the value and the control group; ^b^ indicates a significant difference (*p* < 0.05) between the value and the low-dose iAs^3+^ group. The *p*-value is a measure of the difference between the control group and the experimental group; The *F* test was used to assess the significance of the mean difference between two or more samples. The *F* value represents the ratio of two mean squares (effect item/error item), with no negative value possible. The greater the *F* value, the more obvious the difference, and the higher the test accuracy.

**Table 5 toxics-11-00578-t005:** Effect of PTEN overexpression on sodium arsenite on Notch1/HES1/PTEN and autophagy-related protein expression in fibrosis-causing LX-2 cells.

Group	Notch1	HES1	PTEN	Beclin-1	LC3
Control	0.27 ± 0.05	0.44 ± 0.09	0.99 ± 0.12	0.88 ± 0.04	0.60 ± 0.07
Blank plasmid group	0.19 ± 0.03	0.36 ± 0.05	0.88 ± 0.07	0.76 ± 0.04	0.52 ± 0.05
High-dose iAs^3+^ + blank plasmid group	0.42 ± 0.03 ^a^	0.96 ± 0.04 ^a^	0.66 ± 0.14 ^a^	1.17 ± 0.06 ^a^	1.10 ± 0.18 ^a^
PTEN overexpression group	0.12 ± 0.04 ^b^	0.44 ± 0.02 ^b^	1.26 ± 0.03 ^b^	0.71 ± 0.03 ^b^	0.52 ± 0.08 ^b^
High-dose iAs^3+^ + PTEN overexpression group	0.16 ± 0.03	0.48 ± 0.10	0.87 ± 0.11 ^a^	0.82 ± 0.06	0.54 ± 0.12
*F*	34.569	38.920	13.469	42.223	15.295
*p*	<0.001	<0.001	<0.001	<0.001	<0.001

Note: The values given are relative expression levels from the PCR or western blot analysis. ^a^
*p* < 0.05, compared with the control group; ^b^
*p* < 0.05, compared with the high-dose iAs^3+^ + blank plasmid group. The *p*-value is a measure of the difference between the control group and the experimental group; The *F* test was used to assess the significance of the mean difference between two or more samples. The *F* value represents the ratio of two mean squares (effect item/error item), with no negative value possible. The greater the *F* value, the more obvious the difference, and the higher the test accuracy.

**Table 6 toxics-11-00578-t006:** Effect of PTEN overexpression on LX-2 cell fibrosis caused by sodium arsenite.

Group	α-SMA	Collagen Ⅰ
Control	0.64 ± 0.06	0.09 ± 0.01
Blank plasmid group	0.59 ± 0.02	0.11 ± 0.01
High-dose iAs^3+^ + blank plasmid group	1.02 ± 0.11 ^a^	0.21 ± 0.00 ^a^
PTEN overexpression group	0.63 ± 0.06 ^b^	0.11 ± 0.00 ^b^
High-dose iAs^3+^ + PTEN overexpression group	0.60 ± 0.07	0.15 ± 0.01
*F*	20.115	47.257
*p*	<0.001	<0.001

Note: The values given are relative expression levels from the western blot analysis. ^a^
*p* < 0.05, compared with the control group; ^b^
*p* < 0.05, compared with the high-dose iAs^3+^ + blank plasmid group. The *p*-value is a measure of the difference between the control group and the experimental group; The *F* test was used to assess the significance of the mean difference between two or more samples. The *F* value represents the ratio of two mean squares (effect item/error item), with no negative value possible. The greater the *F* value, the more obvious the difference, and the higher the test accuracy.

## Data Availability

Data from this study may be accessed upon reasonable request from the corresponding author.

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
