# Peer review of "PTEN Overexpression Alters Autophagy Levels and Slows Sodium Arsenite-Induced Hepatic Stellate Cell Fibrosis"

_toxics, 2023, doi:10.3390/toxics11070578_

Round 1
Reviewer 1 Report
Comments:
The present study aims to investigate the effects of PTEN on autophagy levels and arsenite-induced hepatic stellate cell fibrosis by using in vivo and in vitro models. The effect of sodium arsenite exposure on liver histopathological changes in mice was observed using HE and Masson staining. Additionally, autophagosomes in LX-2 cells were observed using a transmission electron microscope in each group.The experimental design, which utilized PTEN overexpression plasmid, Western blot, RT-PCR to determine the levels of protein/mRNA expression in autophagy and liver fibrosis is reasonable. Generally speaking, this study is of some value to elucidate the mechanism of liver fibrosis induced by arsenic. Before publication, I suggest that the following issues be revised accordingly.
1. In Materials and methods section, the concentration of arsenic exposure in LX-2 cells needs to be supplemented, the specific methods of electron microscopy need to be described, and the specific sequence information of PTEN plasmid is recommended to be supplemented.
2. In the Results section, the description of the result is confused between Mice and LX-2 cells study. For example: In 3.2 part of Results section, the subtitle shows “PTEN is involved in the autophagy and fibrosis in the mouse liver tissue caused by sodium arsenite treatment”, however, the results describe the LX-2 cell experiment findings. It is recommended that the author should carefully check and modify this section.
3. The Figures of pathological findings and electron microscopy results are too fuzzy, requiring improved resolution and indication of specific lesion locations.
4. All the Tables are not standardized and lack an indication of units and specific statistical differences (e.g. What does “a” and “b” mean?).
5. The authors also need to pay particular attention to English grammar, spelling, expression, and sentence structure.
The authors also need to pay particular attention to English grammar, spelling, expression, and sentence structure.
Author Response
Dear editor,
Thank you for considering our manuscript for publication in the Toxics journal. We also appreciate the constructive comments from the reviewers, which greatly helped us to improve the manuscript. The manuscript was carefully revised and our point-by-point responses are listed below. We hope that all of the comments have been addressed accurately. The sections of the revised manuscript that have been changed have been highlighted yellow.
We hope you agree that the manuscript is now much more suitable for publication in Toxics. However, if any issues remain, we will be happy to address them.
Yours faithfully,
Insert author name:Fei Huang1, Guanxin Ding1*, YanjieYuan2, Lijun Zhao3,Wenmeng Ding4, Shunhua Wu4*
Reviewer #1
- In Materials and methods section, the concentration of arsenic exposure in LX-2 cells needs to be supplemented, the specific methods of electron microscopy need to be described, and the specific sequence information of PTEN plasmid is recommended to be supplemented.
Response: Thank you, I have added the concentration of arsenic exposure in LX-2 cells(shown in the new Table 1)and the specific methods of electron microscopy have now been included in the materials and methods section(highlighted yellow).
- In the Results section, the description of the result is confused between Mice and LX-2 cells study. For example: In 3.2 part of Results section, the subtitle shows “PTEN is involved in the autophagy and fibrosis in the mouse liver tissue caused by sodium arsenite treatment”, however, the results describe the LX-2 cell experiment findings. It is recommended that the author should carefully check and modify this section.
Response: Thank you for pointing out this mistake. In the Results section, under the subtitle “PTEN is involved in the autophagy and fibrosis in the mouse liver tissue caused by sodium arsenite treatment”, I have changed the wording to make it clear that the mRNA and protein levels were measured in mouse liver samples not LX-2 cells as suggested in the first sentence of that section. The rest of the results section has also been checked and corrected as needed.
- The Figures of pathological findings and electron microscopy results are too fuzzy, requiring improved resolution and indication of specific lesion locations.
Response: Thank you for pointing this out. The figures have been improved as suggested.
- All the Tables are not standardized and lack an indication of units and specific statistical differences (e.g. What does “a” and “b” mean?).
Response: We thank the Reviewer for the comment. The superscript a and b indicate significant differences between the values and either the control or other experimental condition. They are explained in the note below the table. The values are relative protein or mRNA expression levels as calculated from the western blots or PCR analysis, so they don’t have units. We are sorry this was hard to follow. The tables have now been modified to make it clearer and to make the terms used standard between them.
- The authors also need to pay particular attention to English grammar, spelling, expression, and sentence structure.
Response: Thank you for pointing this out. The manuscript has now undergone a full edit by a native English speaking medical writer, so we hope the problems have been resolved.

Reviewer 2 Report
The authors considered a very important topic and targeting PTEN for the management of arsenic exposure. The paper is well written. The Figures 1 -3 are of a good quality.
In any case the paper need to revised:
abstract: if “Background” is a section I suggest to add “Methods”, “Results”, “Conclusions”.
In several parts of the text spaces between words were not considered. I suggest to keep attention to this aspect.
Author Contributions: it is necessary to complete
Institutional Review Board Statement: it is necessary to complete
Informed Consent Statement: it is necessary to complete
Data Availability Statement: it is necessary to complete
Acknowledgments: it is necessary to complete
Conflicts of Interest: it is necessary to complete
References: it is necessary to delete the number in brackets in each references. For example:
1. [1] SINGH R, SINGH S, PARIHAR P, et al. Arsenic contamination, consequences and remediation techniques: a review[J]. Ecotoxicology and environmental safety, 2015, 112: 247-270.
In addition the instructions for authors need to be applied.
Author Response
Dear editor,
Thank you for considering our manuscript for publication in the Toxics journal. We also appreciate the constructive comments from the reviewers, which greatly helped us to improve the manuscript. The manuscript was carefully revised and our point-by-point responses are listed below. We hope that all of the comments have been addressed accurately. The sections of the revised manuscript that have been changed have been highlighted yellow.
We hope you agree that the manuscript is now much more suitable for publication in Toxics. However, if any issues remain, we will be happy to address them.
Yours faithfully,
Insert author name:Fei Huang1, Guanxin Ding1*, YanjieYuan2, Lijun Zhao3,Wenmeng Ding4, Shunhua Wu4*
Reviewer #2:
The authors considered a very important topic and targeting PTEN for the management of arsenic exposure. The paper is well written. The Figures 1 -3 are of a good quality.
In any case the paper need to revised:
abstract: if “Background” is a section I suggest to add “Methods”, “Results”, “Conclusions”.
Response: Thank you, for this suggestion. However, we found that the guidance for Toxics requires an unstructured abstract of less than 200 words. So, we have deleted the headings and shortened the abstract a little.
In several parts of the text spaces between words were not considered. I suggest to keep attention to this aspect.
Response: Thank you, for pointing out these errors, we have made a substantial revision on grammar and spelling. The manuscript has also been checked by a native English speaking medical writer.
Author Contributions: it is necessary to complete
Institutional Review Board Statement: it is necessary to complete
Informed Consent Statement: it is necessary to complete
Data Availability Statement: it is necessary to complete
Acknowledgments: it is necessary to complete
Conflicts of Interest: it is necessary to complete
Response: Thank you for these suggestions, we have made all but one change: The article is about experiments on animals and cells, so there is no requirement for informed consent from human subjects. Institutional Review Board Statement has been attached .
References: it is necessary to delete the number in brackets in each references. For example:
- [1] SINGH R, SINGH S, PARIHAR P, et al. Arsenic contamination, consequences and remediation techniques: a review[J]. Ecotoxicology and environmental safety, 2015, 112: 247-270.
Response: Thank you for these suggestions, we have made all changes.
In addition the instructions for authors need to be applied.
Response: Thank you, we have edited the manuscript to follow the guidance for Toxics. This includes shortening the abstract and editing the reference format.

Reviewer 3 Report
The report is attached.

The report is attached.
Author Response
Dear editor,
Thank you for considering our manuscript for publication in the Toxics journal. We also appreciate the constructive comments from the reviewers, which greatly helped us to improve the manuscript. The manuscript was carefully revised and our point-by-point responses are listed below. We hope that all of the comments have been addressed accurately. The sections of the revised manuscript that have been changed have been highlighted yellow.
We hope you agree that the manuscript is now much more suitable for publication in Toxics. However, if any issues remain, we will be happy to address them.
Yours faithfully,
Insert author name:Fei Huang1, Guanxin Ding1*, YanjieYuan2, Lijun Zhao3,Wenmeng Ding4, Shunhua Wu4*
Reviewer #3
The conclusion is totally absent in the manuscript, and I suggest the authors to add this part (the last part of the abstract, after implementation, should be moved and could be a good starting point for conclusion
section).
Response: Thank you for this suggestion. A conclusion has now been included at the end of the discussion.
This manuscript may have some merits. However, from my point of view, there are also some key-faults.
1) in line 100-101 the authors wrote “The LD50 of sodium arsenite in the mice was 16.2 mg/kg, as determined by the Horn method.” No references were added. It is a key point. Moreover, it is an oral LD50? I consulted
some official/reputable databases and references (e.g. ECHA, PubChem, WHO monograph … and I found different values).
Response: Thank you for raising this important issue. We used the Horn method to find the median lethal dose for mice by ourselves. We did not refer to the IC50 values in other literature, because we think that different mouse types, different valence states of arsenic, different environments, humidity, temperature, and mouse states, will mean the lethal dose is not the same. This may be why you found different values in the literature.
In addition, our experimental article has been published, mainly in Chinese, and could not be written in the reference.(袁艳杰, 黄菲, 日沙来提·塔依尔, 等. 亚砷酸钠暴露小鼠肝纤维化及自噬相关蛋白的表达[J]. 环境与健康杂志, 2021, 38(11):616-620. )
Horn's method is composed of preliminary test and formal test. The preliminary test determines the exposure dose of the test substance in the preliminary test according to the relevant toxicological data of the similar structural chemical substances; the formal test dose series can be determined according to the results of the preliminary test. The animals were randomly divided into 4 groups, 4 animals in each group. The test substance was diluted with distilled water in different doses, and administered orally at 0.01 mL/g. After exposure, the animals were closely observed and recorded in detail for abnormal behavior, the number of animals that died and the time of death. The observation period of all infected animals was 14 days. We have now explained this method and added a reference for the Horn method (ref 24).
2) some sentences are interrupted or unrelated to the context (e.g. line 40)
Response: Thank you for this suggestion. The manuscript has undergone a full edit by a native English speaking medical writer, so we hope these issues have now been resolved.
3) line 51-55 reference missing
Response: Thank you for pointing this out. New references have now been added to this section (refs 13, 15, 18, and 19).
4) M&M - Add (Company, City, Country) for all instrumentations reagents standards, software. Standardize
and change it in all manuscript.
Response: Thank you for this suggestion. Most of this information was in a separate section but this has now been included alongside the first mention of the instrument, reagents, and software.
5) Sodium arsenite is solubilized in deionized water and administered to the animals. So I would like to know
if it is freshly prepared each day? As(III) maybe be oxidized in As(V) in the condition of temperature/humidity reported. As(V) has an LD50 higher than As(III).
Response: Thank you for this suggestion. In order to prevent the oxidation of sodium arsenite, we fed the water to the mice immediately after preparing the water containing sodium arsenite. The mice were only fed water once a day, so that the mice were thirsty when they drink water every day, drink water quickly, and reduce the exposure of water to the air. We have updated the methods to clarify this point.
- line 110 reference missing!
Response: Thank you for pointing this out. The reference has been added for the Horn method (ref 24). The ethics committee reference was: Ethical approval number: IACUC-20210309-08
7) line 117 reference or kit specification should be added
Response: Thank you for pointing this out. We have added the manufacturer (Solarbio, Beijing, China).
8) line 130 PEI? Polyethylenimine?? Please specify.
Response: Yes, it was polyethylenimine. This has now been explained.
9) Cell transfection - Did you use a transfection control? Which?
Response: The transfection control was an empty plasmid for the control group.
- Figure 1: Why is medium-dose group not shown?
Response: Thank you for this question. Because we found that the low-dose group was not significantly different from the control group during the experiment, we removed the low-dose group results from the data. The medium-dose group is the low-dose group after we divided into three groups.
11) line 223 “As shown above, NaAsO2…”, maybe below… but specify where… table 3 I guess.
Response: This has been removed and rewritten during the edit.
12) line 231 Again medium-dose group disappeared. If this group was discarded by the in vivo study, it is necessary to specify what happened and explain the motivations of your choice.
Response: As for our response above, because we found no significant difference between the low-dose group and the control group in our experiment, we removed the low-dose group. The medium dose group was the low dose group and we divided it into three groups.
13) the trend of expression shown in table 3 seems to be in contrast with WB especially for LC3 and Beclin-1
Response: We did find some differences between the protein and mRNA expression levels. We hope that this difference has been explained more clearly after our edit of the results section.
14) please control all the MS. There are many typos (e.g. lines 188, 301, 315 …)
Response: The manuscript has undergone a full edit by a native speaking medical writer.
I suggest authors to improve the discussion. In general data should be shown and explained with a most “transparent manner”. Based on these comments I strongly encourage the authors to improve the manuscript, since in the present form it is not eligible for publication in Toxics.
Response: Thank you for this suggestion, we have edited the discussion to describe the results first and then compare them to the results already presented in the literature. We hope this makes the discussion clearer.

Round 2
Reviewer 3 Report
The authors addressed the most part of reviewer’s comments/suggestions and modified the manuscript accordingly.
However, about the question of the groups the authors wrote in the response to reviewers:
“Response: Thank you for this question. Because we found that the low-dose group was not significantly different from the control group during the experiment, we removed the low-dose group results from the data. The medium-dose group is the low-dose group after we divided into three groups.”
This point is not clear at all in the manuscript and it should be clarified. In the new version of the manuscript, you write (line 86) “The mice were divided into four groups…” and after simply deleted the middle dose group (the “EX”- low-dose group (L), NaAsO2 0.27 mg/kg). So, you present three groups for the animal study.
Cell culture: Similarly, for cell culture you used four groups (control, high, middle, low – Figure 1) and again the middle dose group disappears from results and discussion. Are there no differences both for animal tests and cell cultures?
If you do not clarify this point, some statements, as an example line 361-362 “In this study, the PTEN mRNA and protein levels were downregulated in each iAs3+ dose group”, appear not clear in my opinion.
Probably it is better or, again, more transparent say in the discussion that these groups (1) “animal” middle group and (2) “cell” middle group are not included in the elaborations because “not significantly different from the control group”. However, this implies that the doses chosen for the study and the LD50 were not properly chosen!?!
Minor issue:
Thank you for the explanation of the determination of LD50 by Horn’s method. Unfortunately, I am not able to verify your publication in Chinese (袁艳杰, 黄菲, 日沙来提·塔依尔, 等. 亚砷酸钠暴露小鼠肝纤维化及自噬相关蛋白的表达[J]. 环境与健康杂志, 2021, 38(11):616-620.), but I have to underline that reference you indicated in the manuscript (ref 24 - Wang, L.X. Comparative evaluation of Horn’s method and sequential method in acute oral toxicity test in the new national standard. Agrochemicals 2019, 4, 288–290) is not available. But I guess you used this national standard “National Health and Family Planning Commission of the People’s Republic of China (2014) Acute oral toxicity test (GB 15193.3-2014).”
For reference 24, please add a doi.
Again, I confirm the study has some merits but, due to these points I still have some doubts. So, I have to confirm major revisions and ask/attend the editor’s opinion/decision.
Author Response
Dear editor:
Please below the word

Round 3
Reviewer 3 Report
Congratulations for the work done! I am always happy when my suggestions help the authors improve their manuscript.